# Invertible Hierarchical Generative Model for Images

**Heikki Timonen**                                                                    *heikki.timonen@aalto.fi*
*Department of Computer Science, Aalto University*

**Miika Aittala**
*NVIDIA*

**Jaakko Lehtinen**
*Department of Computer Science, Aalto University*
*NVIDIA*

**Reviewed on OpenReview:** *https: // openreview. net/ forum? id= 4rkKN4tM63*

## Abstract

Normalizing flows (NFs) as generative models enjoy desirable properties such as exact invertibility and exact likelihood evaluation, while being efficient to sample from. These properties, however, come at the cost of heavy restrictions on the architecture. Due to these limitations, modeling multi-modal probability distributions can yield poor results even with low-dimensional data. Additionally, typical flow architectures employed on real image datasets produce samples with visible aliasing artifacts and limited variation. The latent decomposition of flow-models also falls short on that of competing methods, with uneven contribution to a decoded image. In this work we build an invertible generative model using conditional normalizing flows in a hierarchical fashion to circumvent the aforementioned limitations. We show that we can achieve superior sample quality among flow-based models with fewer parameters compared to the state of the art. We demonstrate ability to control individual levels of detail via the latent decomposition of our model. Project source code is available at `https://github.com/timoneh/hflow`.

## 1 Introduction

Generative models for image data have taken large leaps of progress in terms sample quality, interpretability and other performance metrics. Generative Adversarial Networks (GANs) (Goodfellow et al., 2014) were ahead of other types of models in terms of sample quality until only very recently. They, however, optimize a different loss from other types of generative models that operate on maximizing the likelihood (or a bound thereof) of the training data. They also require a separate inference network or optimization procedure for inferring latent variables for real data (Creswell & Bharath, 2018). Recently Denoising Diffusion Probabilistic Models (DDPMs) (Sohl-Dickstein et al., 2015; Ho et al., 2020) have caught up with GANs in terms of sample quality (Dhariwal & Nichol, 2021). In their standard form, however, they suffer from lack of semantic structure of the latent space (Preechakul et al., 2022), and have a trade-off between sample quality and sampling speed due to the iterative nature of the denoising process. Variational Autoencoders (VAEs) (Kingma & Welling, 2013; Rezende et al., 2014) have a robust inference procedure with a smooth and semantically meaningful latent space and allow one to retrieve an encoded image with a reasonable reconstruction error as well as compute a lower bound for data likelihood. VAEs can, however, be notoriously difficult to train with the goal of optimizing sample quality and easily give overly smooth and blurry samples or alternatively exhibit poor latent structure (Higgins et al., 2016). Recent work on VAEs has moved the focus on very deep VAEs (Vahdat & Kautz, 2020; Child, 2020) in the search for data-likelihoods that exceed autoregressive models (Van den Oord et al., 2016), whose depth scales linearly with the data-dimensionality. Normalizing flows (Tabak & Turner, 2013) offer exact and fast inference as well as exact likelihood instead of a bound offered by VAEs and DDPMs. Flow-models map data into a latent space deterministically and invertibly,

Ours →
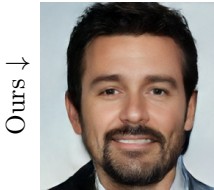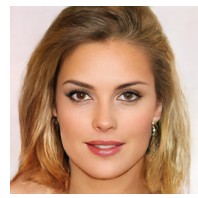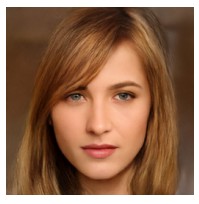
Glow →
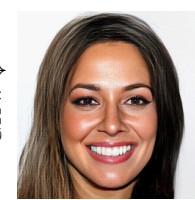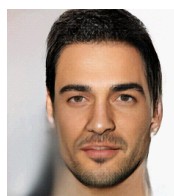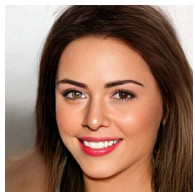

Figure 1: Curated $256 \times 256$ samples from our model (left) trained with CelebA-HQ. Samples from Glow (Kingma & Dhariwal, 2018) (right) are generated using the official implementation and pre-trained weights. Our model produces samples of significantly higher quality in terms of sample variance and spatial consistency, but at much lower model capacity. The reader is encouraged to zoom for a detailed view.

which sets heavy restrictions on their architectures. Enforced invertibility is problematic also for modeling multi-modal distributions (Cornish et al., 2020) due to the topology-preserving nature of NFs. Lastly, Flow-models tend to focus on local pixel correlations while optimizing the data likelihood and disregard the semantic content of an image (Kirichenko et al., 2020).

In this work, we study the performance problems of normalizing flows purely from the perspective of sample quality with image datasets. Past work has mostly focused on finding more expressive flow-architectures to improve a test-set likelihood-score (Dinh et al., 2016; Ho et al., 2019; Hoogeboom et al., 2019; Behrmann et al., 2019; Chen et al., 2019). Despite yielding competitive likelihood scores, pure flow-models yield relatively poor samples compared with other likelihood-based generative models. Even very deep flows fail produce spatially consistent images and lack variation due to heavy truncation, as seen in Figure 1 featuring samples from Glow (Kingma & Dhariwal, 2018). Here we investigate whether we can construct an invertible generative model yet avoid the issues arising from enforced one-to-one invertibility. We show that we can achieve good sample quality with high-resolution natural images with a model that replaces depth with width.

Multi-resolution coarse-to-fine processing is a key principle behind many generative models. While pure flow models such as Glow do employ a resolution hierarchy, we find that latents from different resolutions fail to impact the detail of the associated scale in the expected way. Furthermore, we hypothesize that the invertible squeeze-and-split image resizing operations employed by Glow are not ideally suited for image generation. Indeed, from an image processing perspective they can be likened to highly aliased filters, known to bias the generation towards regular grid and checkerboard artifacts (Karras et al., 2021) such as those apparent in Figure 1 (right) when viewed at high zoom. Conversely, such grid artifacts may be difficult to smooth out using the invertible scale-and-shift convolution layers. These observations and the general difficulties of flow models motivate us to introduce a flexible multi-scale model that employs individual shallow Glow models where needed. We lose the ability to evaluate the exact likelihood, but retain the exact invertibility through a pair of encoder and decoder pipelines.

Inspired by the success of image super-resolution problem with *conditional* normalizing flows (Lugmayr et al., 2020; Liang et al., 2021), we construct a hierarchical stack of shallow Glow-like flows that models different levels of detail via a conditioning mechanism. An image is encoded into abstract decreasing-resolution features by a sequence of general-purpose (non-invertible) networks, discarding detail at each step. Conversely, an image is generated by rebuilding this detail using a sequence of separate Glow-like models, interleaved with general-purpose CNNs that process and upsample the conditioning signal. The system is jointly trained on the weighted sum of the conditional flow-losses, inducing the encoder and decoder to find the appropriate decomposition.

We observe a significant increase in sample quality when compared with deep normalizing flow models, decreasing the Fréchet Inception Distance (FID) metric (Heusel et al., 2017) of Glow 51.5 to 27.3 on the CelebA-HQ-dataset (Karras et al., 2017) at $256 \times 256$ resolution, using a significantly smaller model with a much shorter training time. While there are flow-models with better likelihood-scores, we choose Glow as a reference since few other flow-models have been shown to generate significantly better samples at $256 \times 256$-resolution. For instance, while Residual Flows of Chen et al. (2019) achieve a better likelihood than Glow,

the samples are of very similar quality in terms of subjective visual quality and the FID. We discover a hierarchical latent structure where we can control elements of various levels-of-detail separately. We show that images generated by interpolations in the latent space are smooth and remain close to the manifold of real images. Finally, we notice that individual parts of our model do not need to be very deep and instead we can trade model depth for width.

**Our contribution** In summary, in this work we show that we can construct an invertible model that is trained only with the normalizing flow maximum likelihood-loss and has superior sample quality with $256 \times 256$-resolution CelebA-HQ compared against Glow while being much faster to train. Our model has a smooth latent space and allows both fast sampling and inference of latent variables.

## 2 Method

We first introduce the general idea of a hierarchical conditional normalizing flow (Section 2.1) and associated practical design choices (Section 2.2). We then introduce our architecture by generalizing the idea to a multi-scale hierarchy, and discuss connections to related methods (Section 2.3).

### 2.1 Hierarchical Conditional Normalizing Flows

A normalizing flow $f$ yields an exact value for the model probability density function via the change of variables formula

$$\log p(\boldsymbol{x}) = \log p_{\text{base}}(f(\boldsymbol{x})) + \log |\det \mathrm{d}f(\boldsymbol{x})/\mathrm{d}\boldsymbol{x}|, \tag{1}$$

where $\mathrm{d}f(\boldsymbol{x})/\mathrm{d}\boldsymbol{x}$ is the Jacobian of $f$ with respect to $\boldsymbol{x}$. The function $f$ is constructed to be efficiently invertible and to have a Jacobian determinant with a computational time-complexity preferably linear or better with respect to the data dimensionality. $p_{\text{base}}$ is usually set to a multivariate unit Gaussian distribution. In this work, we follow the architectural choices of NFs presented in Dinh et al. (2016); Kingma & Dhariwal (2018) consisting of a sequence of affine coupling layers, invertible $1 \times 1$ convolutions and spatially broadcast learnable scales and biases with data-dependent initialization "actnorms". We introduce conditioning to a flow by giving an additional input $\boldsymbol{c}$ to the affine coupling layers $h$, which invertibly transform a variable $\boldsymbol{x}$ via

$$h(\boldsymbol{x}; \boldsymbol{c}) = [\boldsymbol{x}_a, \boldsymbol{x}_b \odot \mathrm{NN}_s(\boldsymbol{x}_a; \boldsymbol{c}) + \mathrm{NN}_b(\boldsymbol{x}_a; \boldsymbol{c})], \tag{2}$$

where $\boldsymbol{x}_a$, $\boldsymbol{x}_b$ is some split of the input tensor $\boldsymbol{x}$ and $\mathrm{NN}_s$ and $\mathrm{NN}_b$ are neural networks.

Due to the innate limitations of normalizing flows we want to offload as much work as possible from the invertible neural networks. Inspired by solving the image super-resolution task with flow-models, we train a conditional flow-model in such a way that the conditioning input $\boldsymbol{c}$ is in itself function of the data, and jointly learned in the process of maximizing the likelihood. Essentially this forms an autoencoder whose "reconstruction loss" is the flow loss. The training loss for such a network is given by:

$$L_{\text{cond}}(\boldsymbol{x}) = -\log p_{\theta}(\boldsymbol{x}|\boldsymbol{y}), \tag{3}$$

where $p_{\theta}$ is the distribution induced by a conditional normalizing flow

$$\boldsymbol{x} = f_{\text{cond}}(\boldsymbol{z}_{\text{cond}}; \boldsymbol{y}, \theta), \tag{4}$$

with a unit Gaussian prior. More precisely, $p_{\theta}$ is the density induced by the push-forward $P_{\boldsymbol{X}} = f_{\text{cond}} \# P_{\boldsymbol{Z}_{\text{cond}}}$ of the Gaussian prior $P_{\boldsymbol{Z}_{\text{cond}}}$. $\boldsymbol{y} \sim q_{\phi}(\boldsymbol{y}|\boldsymbol{x})$ is a stochastic encoder with learned components (convolutions) with parameters $\phi$. The encoder is trained jointly with the normalizing flow. This learning objective leads into finding the latent variable representation $\boldsymbol{Y}$ which has maximal mutual information $I(\boldsymbol{X}, \boldsymbol{Y})$ with the original data. If the complex conditional normalizing flow parametrizing $p_{\theta}$ was reduced into a Gaussian likelihood conditioned by $\boldsymbol{y}$ via the mean and covariance, we would recover the standard VAE reconstruction-loss. Like in a VAE, for generation we need to additionally model the "prior" distribution

of $\boldsymbol{Y}$. However, unlike in a standard VAE, we do not directly enforce a Gaussian prior for each $q(\boldsymbol{y}|\boldsymbol{x})$. Instead, we approximate the aggregated posterior in a separate step with a distribution induced by another flow-model

$$\boldsymbol{y} = f_{\text{prior}}(\boldsymbol{z}_{\text{prior}}; \varphi), \tag{5}$$

of density $p_\varphi$ of the respective push-forward between $\boldsymbol{Z}_{\text{prior}}$ and $\boldsymbol{Y}_{\text{prior}}$, using the flow-loss of Equation 1

$$L_{\text{prior}}(\boldsymbol{y}) = -\log p_\varphi(\boldsymbol{y}), \tag{6}$$

where $\boldsymbol{y}$ is sampled via $\boldsymbol{x} \sim p_{\text{data}}(\boldsymbol{x})$, $\boldsymbol{y} \sim q(\boldsymbol{y}|\boldsymbol{x})$. Note that the "prior" label here denotes that the flow $f_{\text{prior}}$ operates on $\boldsymbol{y}$ and is different from the unit Gaussian base distribution of a flow introduced in Equation 1. In practice, we minimize the following expression

$$\min_{\theta, \phi, \varphi} \mathbb{E}_{\boldsymbol{x} \sim p_{\text{data}}(\boldsymbol{x}), \boldsymbol{y} \sim q_\phi(\boldsymbol{y}|\boldsymbol{x})} \left[ L_{\text{cond}}(\boldsymbol{x}) + L_{\text{prior}}(\text{SG}[\boldsymbol{y}]) \right] \tag{7}$$

where $\text{SG}[\dots]$ is the stop-gradient operator with $\text{SG}(\boldsymbol{x}) = \boldsymbol{x}$ but which has vanishing partial derivatives; we discuss the reason for stopping the gradients below. Alternatively, this can be thought of as a two-step training procedure, with the stop-gradient operator decoupling the prior-flow from rest of the model. Figure 2 illustrates the model setup, as well as shows the two-step -nature. From now on, we omit the flow and encoder parameters $\theta, \phi, \varphi$ from the notation for clarity.

**Stop-gradient operations** The loss resembles that used in VAEs, only lacking the negative entropy term of the encoder in the KL-divergence between the approximate posterior and the prior. This introduces a loophole where—in the case of a deterministic encoder—the prior flow loss can be arbitrarily improved by concentrating the distribution of $\boldsymbol{y}$'s, leading to a degenerate solution. A related form of this degenerate solution with vanishing variance was described by Hoffman et al. (2017) in the context of the $\beta$-VAE (Higgins et al., 2016) and an implicit use of a data-dependent prior. Xiao et al. (2019) also described similar degeneracy in their closely related work. For stochastic encoders with Gaussian noise (as used by us; described later), a related loophole

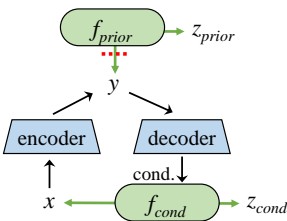

Figure 2: Hierarchical conditional normalizing flow structure. The input $\boldsymbol{x}$ is encoded into the latent $[\boldsymbol{z}_{\text{cond}}, \boldsymbol{z}_{\text{prior}}]$ by general-purpose CNNs (blue blocks) and NFs (green blocks) via the intermediate $\boldsymbol{y}$. The stop-gradient operation is denoted by the red dashed line.

exists through intentionally decreasing the signal-to-noise ratio of $\boldsymbol{y}$, making it uninformative akin to posterior collapse in VAEs. We employ the stop-gradient operations to prevent the prior flow $f_{\text{prior}}$ from directly impacting the distribution of $\boldsymbol{y}$ during training. Adding the entropy-terms into the loss (and bringing the loss even closer to the VAE-loss) can make the tendency for reaching degenerate state even stronger as it directly encourages $q(\boldsymbol{y}|\boldsymbol{x})$ to have an entropy matching that of the prior.

**Separation of modeling tasks** The most important design point of the described model is the choice of the form of the stochastic encoder $q(\boldsymbol{y}|\boldsymbol{x})$ subject to the limitations of the normalizing flows. A deterministic $q(\boldsymbol{y}|\boldsymbol{x}) = \delta_{\boldsymbol{x}}(\boldsymbol{y})$ (the identity) only pushes the modeling work to the flow modeling the prior, $f_{\text{prior}}$, while too strong a bottleneck (either by strong noise or aggressive blurring) does not provide enough conditioning information for the conditional flow. We deliberately avoid using squeeze-and-split operations in both the conditional flow and the prior-flow. That is, none of the operations within a flow change the spatial resolution from that of the respective flow's input. Instead, we design the stochastic encoder $q$ to greatly reduce the spatial resolution with non-aliasing filters, but also allow for learned components (convolutions). Ideally, all perceptually meaningful, spatially long-range correlations that the conditional flow might not be able to consistently capture can be routed via the encoder $q(\boldsymbol{y}|\boldsymbol{x})$. Finally, the resulting distribution of $\boldsymbol{y}$ should be such that it can be approximated — without squeeze-and-split-operations — with another flow.

---

**Algorithm 1** Inference and sampling for the model in Figure 2

---

**Require:** Data point $\boldsymbol{x}$, noise scale $\alpha$
  **procedure** INFERENCE
    $\boldsymbol{y} \leftarrow \text{Encoder}(\boldsymbol{x})$
    $\sigma \leftarrow \sigma_{\boldsymbol{X}}$         $\triangleright$ Defined in Eq. 10
    $\boldsymbol{\varepsilon} \sim \mathcal{N}(\mathbf{0}, \alpha^2 \sigma^2 \boldsymbol{I})$
    $\boldsymbol{y} \leftarrow \boldsymbol{y} + \boldsymbol{\varepsilon}$
    $\boldsymbol{z}_{\text{prior}} \leftarrow f_{\text{prior}}^{-1}(\boldsymbol{y})$
    $\boldsymbol{z}_{\text{cond}} \leftarrow f_{\text{cond}}^{-1}(\boldsymbol{x}; \text{Decoder}(\boldsymbol{y}))$
    $\boldsymbol{z} \leftarrow [\boldsymbol{z}_{\text{prior}}, \boldsymbol{z}_{\text{cond}}]$
    **return** $\boldsymbol{z}$
  **end procedure**

**Require:** Sampling standard deviation $\sigma_{\text{sampling}}$
  **procedure** SAMPLING
    $\boldsymbol{z}_{\text{prior}} \sim \mathcal{N}(\mathbf{0}, \sigma_{\text{sampling}}^2 \boldsymbol{I})$
    $\boldsymbol{z}_{\text{cond}} \sim \mathcal{N}(\mathbf{0}, \sigma_{\text{sampling}}^2 \boldsymbol{I})$
    $\boldsymbol{y} \leftarrow f_{\text{prior}}(\boldsymbol{z}_{\text{prior}})$
    $\boldsymbol{x} \leftarrow f_{\text{cond}}(\boldsymbol{z}_{\text{cond}}; \text{Decoder}(\boldsymbol{y}))$
    **return** $\boldsymbol{x}$
  **end procedure**

---

In the limit of the encoder being only fixed deterministic filtering and downsampling we recover the image super-resolution task with a learned prior on the low-resolution images. The purpose of the *learned* encoder-components is to allow for finding a more compact representation for $\boldsymbol{y}$ than a merely downsampled image. Conversely, we want to avoid near-perfect auto-encoding via the $\boldsymbol{y}$-variable. We require the conditional normalizing flow to be able to generate an appropriate level of spatially coherent detail, such that the capacity of the conditioning $\boldsymbol{y}$-variable be used to only model global, high-level features. Otherwise, the capacity of the conditional flow to produce variation will essentially be wasted if each image can be generated using only a higher-level representation.

**Invertibility and sampling**   Our construction remains invertible as each datapoint can be encoded into a latent with

$$\boldsymbol{y} \sim q(\boldsymbol{y}|\boldsymbol{x}) \tag{8}$$

$$\boldsymbol{z} = [\boldsymbol{z}_{\text{prior}}, \boldsymbol{z}_{\text{cond}}] = \left[ f_{\text{prior}}^{-1}(\boldsymbol{y}), f_{\text{cond}}^{-1}(\boldsymbol{x}; \boldsymbol{y}) \right], \tag{9}$$

that is, *the datapoint is lifted into a space that has more dimensions than the original datapoint.* Any encoded datapoint can deterministically be recovered with zero reconstruction-error via Equations 4 and 5, since $f_{\text{prior}}$ and $f_{\text{cond}}$ are by construction bijective. This holds for any type of encoder $q(\boldsymbol{y}|\boldsymbol{x})$. The inference and sampling procedures for the model in Figure 2 are summarized in Algorithm 1.

## 2.2 Individual design choices

**Stochastic encoders**   While the encoder $q$ could be deterministic, we empirically find it useful to inject Gaussian noise to the outputs of the encoders. That is, we have $q(\boldsymbol{y}|\boldsymbol{x}) = \mathcal{N}(\boldsymbol{y}; \mu(\boldsymbol{x}), \alpha^2 \sigma_{\boldsymbol{X}}^2 \boldsymbol{I})$, where

$$\sigma_{\boldsymbol{X}} = 1/\dim \boldsymbol{y} \left( \sqrt{\text{Var}(\mu(\boldsymbol{X}))}^{\text{T}} \mathbf{1} \right), \tag{10}$$

where $\mu$ is parametrized as a neural network (CNN with downsampling in Figure 3, the encoder in Figure 2). We choose the $\sigma$ of the noise to be proportional to the variance (with parameter $\alpha$) to prevent the model from scaling the signal up to improve signal-to-noise ratio and effectively ignoring the noise. The mean is taken in order to have isotropic noise so that the model cannot corrupt only a part of elements of $\boldsymbol{x}$ and squeeze an almost clean signal in the remaining elements. Assumptions of similar spirit of data corrupted with isotropic (instead of anisotropic) noise have been found to yield higher-quality samples within the VAE-literature (Rybkin et al., 2021). In practice, the variance is computed as a Monte Carlo estimate over the current minibatch by computing $\mu$, taking the empirical element-wise standard deviation over the minibatch and normalizing with the dimensionality of $\boldsymbol{y}$.

The noise can be thought of as a regularizer or as augmentation like in the work of Ghosh et al. (2019). An alternative view is to consider the noise as mollification for the prior distribution, rendering it more

reasonable to model with the prior-flow. The purpose of the noise is hence not the same as in VAEs (we could choose to have a deterministic encoder), where it is strictly a construction for allowing optimization of the Evidence Lower Bound (ELBO). Here, $\sigma$ of the Gaussian $q(\boldsymbol{y}|\boldsymbol{x})$ does not appear in the loss and hence *our optimization target is not a bound for the data-likelihood*, missing the entropy-term of the KL-divergence of the VAE-loss. We can, however, compute a Monte Carlo estimate of the ELBO by adding the missing entropy-term, which can readily be computed for the Gaussian-conditional stochastic encoder.

**Multi-scale architecture** We introduce a multi-scale representation for $\boldsymbol{y}$ to enforce more hierarchy into the generation process. That is, instead of encoding $\boldsymbol{x}$ into a single $\boldsymbol{y}$ (Figure 2), we create a multi-scale representation $\boldsymbol{y} = [\boldsymbol{y}_{R_1}, \boldsymbol{y}_{R_2}, \ldots, \boldsymbol{y}_1, \boldsymbol{y}_{\text{prior}}]$, where $R_i$ are the resolutions of the $\boldsymbol{y}$-variables up to $\boldsymbol{y}_1$ and $\boldsymbol{x} = \boldsymbol{y}_{R_1}$. The hierarchical encoder–decoder structure is illustrated in Figure 3. The conditional part of loss of the training-loss now becomes

$$L_{\text{cond}}(\boldsymbol{x}) = - \sum_{R_i \in \mathcal{R}} \log p(\boldsymbol{y}_{R_i}|\boldsymbol{y}_{<R_i})$$
$$- \log p(\boldsymbol{y}_1|\boldsymbol{y}_{\text{prior}}), \tag{11}$$

where each $p$ is induced by a separate conditional normalizing flow. In training, we weight each term in Equation 11 by a separate weight $w_i$. Sampling follows the same procedure as in the two-level model of Figure 2, with independent Gaussian samples drawn from the base distribution of each of the flows followed by their conditional inversion using the results from the higher levels of hierarchy.

The multi-level hierarchical framework is in fact quite general and does not enforce any particular form for the normalizing flows used to model conditional distributions of Equation 11. One could also use probabilistic models of other types instead of normalizing flows. However, with other types of model, one may lose strict invertibility or suffer in terms of sampling efficiency. For example, Preechakul et al. (2022) introduce a two-stage model which can be thought of as a similar hierarchical construction but with normalizing flows replaced with denoising diffusion models. We choose to remain close to the Glow-architecture with the choice of model for the conditional distribution to highlight the benefits of replacing the limited resolution-hierarchy of the baseline Glow-model with our proposed hierarchical construction. Modifications are justified with ablation studies in Section 3.2.

**Trading depth for width** We drastically reduce the number of latent variables when compared with deep VAEs, by only using $\sim$ 5 levels for $\boldsymbol{y}$. Deep VAEs have an order of magnitude more depth (Child, 2020; Vahdat & Kautz, 2020). Instead, we move capacity to the normalizing flows (which themselves are shallow compared to Glow) and ensure that each flow-prior models meaningful variation in the data, instead of merely adding Gaussian noise to the output of the previous decoder. There has

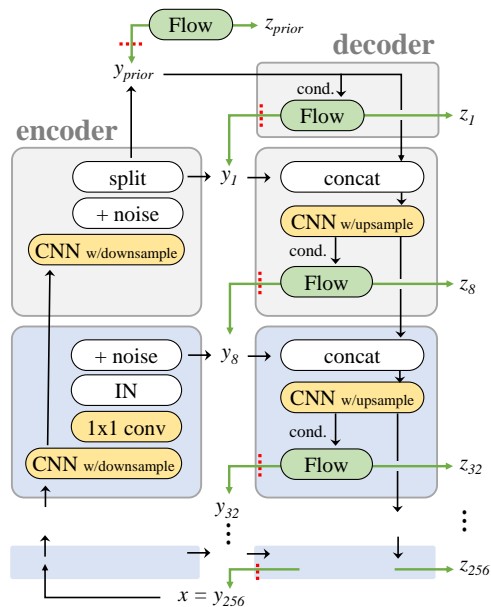

Figure 3: Multi-scale architecture. The encoder (left) and decoder (right) pipelines consist of repeating per-resolution blocks (light blue) with each intermediate representation $\boldsymbol{y}_{R_i}$ further encoded into Gaussianized latents $\boldsymbol{z}_{R_i}$ via normalizing flows (green). Refer to Appendix F for a detailed breakdown of the CNN and flow layers.

been evidence in previous work that replacing depth in VAEs with the ability to handle long distance interdependencies with attention within the latent code yields competitive likelihood-scores (Apostolopoulou et al., 2021). However, it is not clear if this also translates into better image quality in high-resolution images.

**U-Nets in Normalizing Flows**   We find it necessary to use U-Nets (Ronneberger et al., 2015) in the coupling layers of the normalizing flows despite the desire for routing semantic information via a lower-resolution latent. Low-level detail that is not encoded into the latent might still have spatially long-range dependencies (long strands of hair, color of the eyes) and hence equipping the flows with tools to model these dependencies is required. If the conditional normalizing flows operate only on very local detail, the model has to route these low-level features via the high-level latent, potentially stealing capacity from other useful high-level features and violating our design principles. Bottle-necking the encoder–decoder pipeline with e.g. blurring might also render this impossible and the aforementioned details might simply be lost. Within flows using U-Nets, we split the tensor spatially in $8 \times 8$, $4 \times 4$ and $2 \times 2$ checkerboard-patterns to differentiate from the Glow-like split in the channel-direction. While an equivalent split can be achieved with the spatial-to-channels squeeze operations and pure channel-direction splits, we do not observe Glow-like grid artifacts in any of our samples.

## 2.3   Related Work

**VAEs**   Our model can be seen as a special case of a VAE, where instead of a Gaussian likelihood we have more complex normalizing flow. Our approach also models the aggregated posterior $\mathbb{E}_{\boldsymbol{x} \sim p_{\text{data}}(\boldsymbol{x})} \left[ q(\boldsymbol{y}_{\text{prior}} | \boldsymbol{x}) \right]$ and disregards the negative entropy-term of the approximate posterior–prior type KL-divergence terms in the optimization process. There is no pressure from the perspective of the optimization loss to have each $q(\boldsymbol{y} | \boldsymbol{x})$ be zero-mean unit Gaussian, preventing the phenomenon known as "posterior-collapse", "information preference" or "optimization challenges" of VAEs (Bowman et al., 2015; Chen et al., 2016). We see the addition of noise more as a regularization technique rather than a necessity for a variational bound. Even more generally, the difference between VAEs and NFs is not always very clear. A flow-model trained on noise-augmented data can be seen as a VAE and vice versa (Huang et al., 2020). Some VAEs also use normalizing flows as components for a more expressive posterior approximation or likelihood-model (Rezende & Mohamed, 2015; Kingma et al., 2016; Agrawal & Dukkipati, 2016). Finally, while normalizing flows are usually described as being able to yield exact likelihoods, they often employ "dequantization" to render discrete data continuous. However, dequantization, or the addition of uniform (Theis et al., 2015) or more complex noise (Ho et al., 2019) to lift the data distribution to a continuous space renders a model to only give bounds of likelihoods rather than an exact likelihood.

**Flows with noise-augmented data**   Huang et al. (2020); Chen et al. (2020); Grcić et al. (2021) all share the same idea of lifting the data distribution into a higher-dimensional space via padding of noise for fixing issues with multi-modal data and invertibility with normalizing flows. Their work focuses mostly on improvements in the likelihood score and for images with resolutions less than $256 \times 256$. Our construction of the padding is more delicate. Rather than padding the original data with noise, we pad the data with a slightly noisy (or even noiseless), maximally informative compression of the data which is specifically designed to combat issues of flow-models on images.

**Wavelet Flow**   Yu et al. (2020) build a similar hierarchy of conditional flows, but with direct one-to-one invertibility and exact likelihood-evaluation using Haar-wavelet transforms as a fully invertible encoder–decoder structure. The conditional flows generate the the detail coefficients conditioned on the mean. Compared with their work, we see benefit in using an overcomplete representation to allow a more informative high-level latent (that is, one that is not necessarily a low-resolutions image). We also avoid aliasing due to not using box-filtering. We do however, lose the ability to measure exact likelihood.

**Flows on manifolds**   Instead of defining data-likelihood for the space of all RGB-images of a fixed resolution, one can choose to parametrize a manifold using points in a lower-dimensional space and model density only on this manifold. As an image is unlikely to lie exactly on the manifold, one needs to work with the projections of images onto the manifold. The training process hence comprises of two steps: finding the manifold that is on average closest to the training data, and modeling the density of the projected data on the manifold. Several pieces of prior work explore ways of achieving this: Kothari et al. (2021); Brehmer & Cranmer (2020). Our work is in spirit very similar, yet we do not limit the density modeling to a manifold but work in the full space of RGB-images. In particular, we do not simply train a deterministic autoencoder

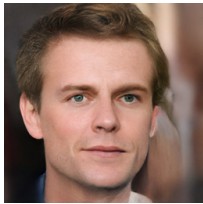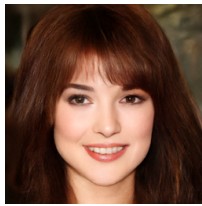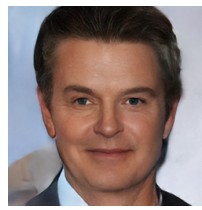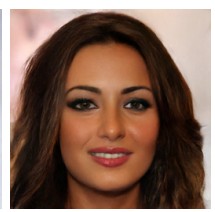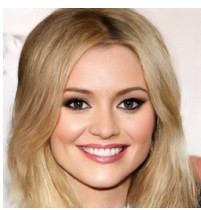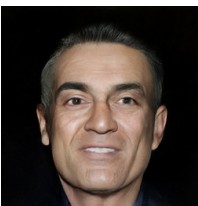

Figure 4: Curated samples from our model (Config A) trained with CelebA-HQ $256 \times 256$ using truncated $\sigma_{\text{sampling}} = 0.7$ (reduced-temperature sampling as in Kingma & Dhariwal (2018)) for latent resolutions greater than 1.

to approximate the manifold of real images and then fit a flow-model into the latent space of the autoencoder. Our encoders and decoders are trained *jointly* with the conditional flows, using their likelihood-loss as the training signal.

## 3   Results

Our model greatly improves the FID for Flow-based models using the CelebA-HQ (Karras et al., 2017) dataset in $256 \times 256$: our model reaches a score of 27.3 against Glow's 51.5 with only about 36% of Glow's parameters [1]. Furthermore, we measure a throughput of around 50 samples / second on an NVIDIA RTX 3090 GPU, which is around 4 times the throughput of Glow on the same hardware. Next, we show qualitatively that our model constructs a latent decomposition which allows controlling individual levels of detail independently and in a more uniform fashion than Glow. We also present ablation studies, showing how different architectural choices within our model change its behavior and performance in terms of FID. Finally, we train baseline Glow-like flow-models with similar capacity to ours using the church and bedroom classes of the LSUN-dataset (Yu et al., 2015). We present a simple toy-case on a 2D mixture of Gaussians, showing the ability of our model to capture multimodal distributions, in Appendix E. Likelihood-scores of the models are tabulated in Appendix G. Details on the training parameters are aggregated in Appendix H.

### 3.1   Qualitative model behavior

Samples from a model trained with CelebA-HQ $256 \times 256$ exhibit good spatial coherency and variation both in high and low-level detail (Figure 4). Samples from our model also lack checkerboard-like aliasing artifacts that can be seen in the Glow counterparts.

Though the effect of $z_{256}$ is subtle, each part of the latent has an observable effect on the decoded image, which is not true for Glow. Figure 5 shows pixel-wise standard deviations for an encoded image when sampling a part of the full latent from the prior but fixing the others. We perform the same operation for Glow. With our model, the latent codes of increasing resolution change increasingly high-frequency details in the image. For example $z_1$ changes the identity of the person and the background, while $z_{128}$ mostly affects the fine structure of the hair. Compared with Glow, our latent structure has a more uniform effect on a decoded image, with the high-resolution latent codes also yielding visible changes in the image. We encourage the reader to also look at Video 3 from the supplementary material for another visualization of the variance.

Figure 6 presents another view to the latent decomposition by showing how various levels of detail are encoded into the latent space. We encode a real image, and cumulatively set $z_{R_i}$ to zero starting from high resolutions. This process gradually removes detail from the image with the $z_{\text{prior}}$ and $z_1$-variables only containing very high-level information like the orientation of the head and the hair color. Interestingly, the eye-color is stored in $z_{32}$, showing that being able to model long-range correlations within the normalizing flows is still required despite the encoder–decoder -mechanism within our model. Otherwise, there would be inconsistencies in the samples of our model, like mismatching eyes.

---

[1]We computed the parameter count of Glow using the official pre-trained model.

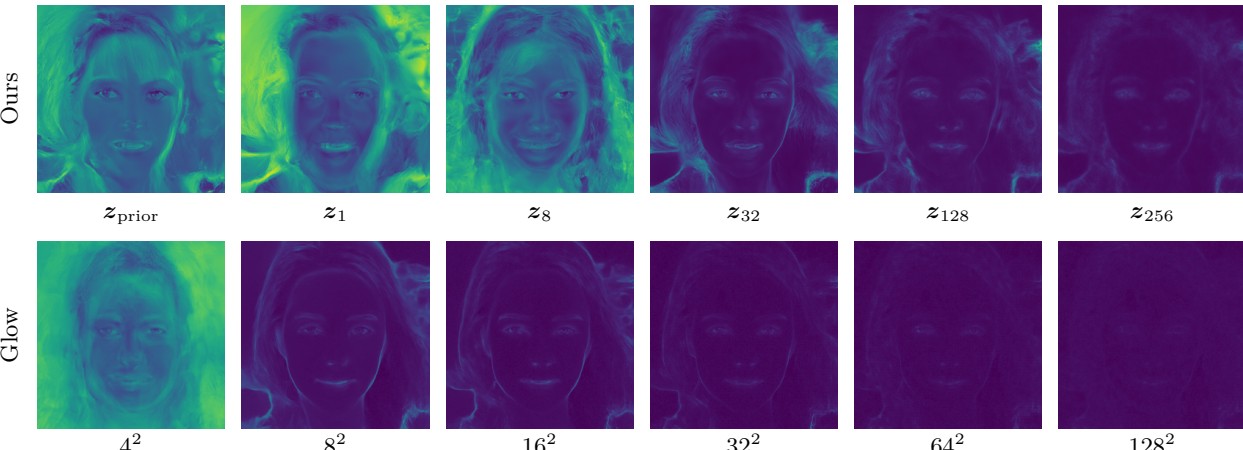

Figure 5: Pixel-wise standard deviations while re-sampling of latents from the prior one resolution at a time (columns) with 32 samples and no truncation. The latent code of each resolution is responsible for changing detail of the corresponding level of detail. Compared with Glow, our method yields a more uniform effect of latents of different resolutions. *Note that the spatial dimensions of the latents differ between the corresponding colums.*

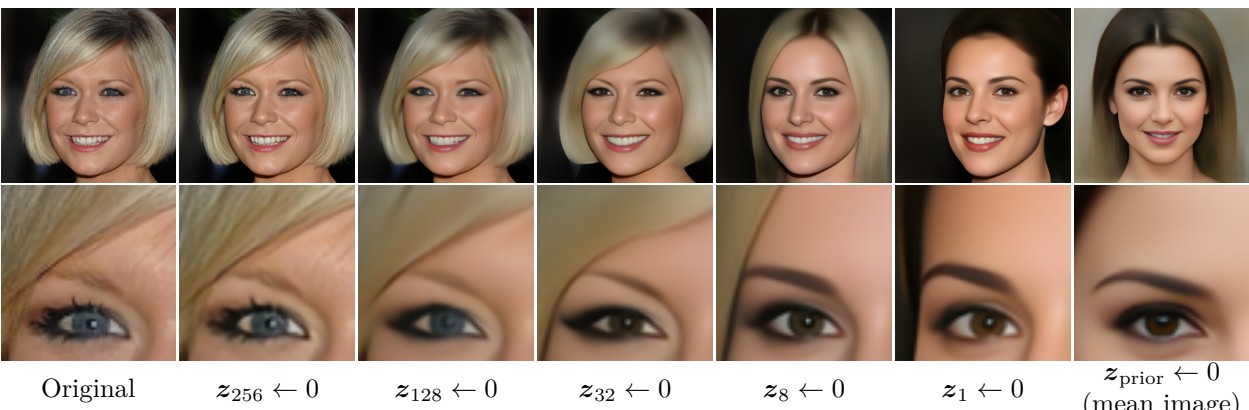

Figure 6: To complement Figure 5, we visualize the contributions of individual latents by inverting a real image (left) and cumulatively zeroing them starting from the finest resolution. This causes a progressive loss of detail at larger and larger scales. The rightmost image is the the result of setting *all* latents to zero.

Interpolations in the latent space of our model result in smooth changes in the decoded images, but also yield sharp images. In contrast to Glow, our interpolations also lack strong aliasing-artifacts. Videos on interpolations in the latent space for random samples and real images can be found in the supplementary material.

## 3.2 Ablations

We train our model using several variations in the configuration. We identify important design-elements that affect sample quality, compared by FID-values in Figure 7a. We refer the reader to Appendix A for a supporting visualization for the ablations, similar to Figure 6.

**Noise in Encoders** From Figure 7a we notice that adding a non-zero amount noise to the $\boldsymbol{y}$-variables is beneficial in terms of sample quality. If no noise is added (Config B) — rendering the encoder deterministic — we notice that the model tries to encode an increasing amount of low-level detail into the high-level, low-resolution latents. Conversely, for a high amount of noise (Config C), images after the aforementioned procedure become increasingly blurry. Hence we need to specifically tune the noise-level for optimal sample-quality. We hypothesize that the added noise is more destructive to high-frequency details of a $\boldsymbol{y}$-variable and renders it more favorable to encode global features into higher-level $\boldsymbol{y}$-variables. The decoder can tolerate noisy $\boldsymbol{y}$ inputs to a limit, but too heavy noise likely starts to degrade the results, causing blurring. In the limit of very strong noise, the signal of $\boldsymbol{y}_{R_i}$ is lost and the task of the respective normalizing flow becomes trivial (a "posterior collapse"), due to $\boldsymbol{y}_{R_i}$ already being almost Gaussian.

**U-Nets in Normalizing Flows** Employing U-Nets in the normalizing flows is beneficial in terms of FID. A model with no U-Nets in the normalizing flows (Config D) has a stronger preference to attempt to encode this into the high-level latent code improve the flow-losses. We attribute the this failure to the disability of the model to generate this content using the high-resolution $\boldsymbol{y}$ modeling normalizing flows. With only a limited receptive field the model has to use the capacity of the high-level latent for these features. We also see traces of very low-frequency noise which may be an attempt of the model to represent the low frequencies of the slight Gaussian noise used in data augmentation via the low-level latents.

**Flows at additional resolutions** In the best-performing Config A, we do not take constant-size steps down in resolution within the decoder but decompose $\boldsymbol{y} = [\boldsymbol{y}_{256}, \boldsymbol{y}_{128}, \boldsymbol{y}_{32}, \boldsymbol{y}_8, \boldsymbol{y}_1, \boldsymbol{y}_{\text{prior}}]$. Surprisingly, adding additional resolutions levels to the encoder–decoder (Config E) to model the missing resolutions $\boldsymbol{y}_{64}$, and $\boldsymbol{y}_{16}$, (while reducing capacity at other flows, encoders and decoders, to have an approximately similar number of parameters), renders the results considerably worse. We again observe that high-frequency details are again encoded more aggressively into the low-resolution latents. While the model might have lower-capacity encoders and decoders, it is clearly misusing the given resources.

**Deeper Normalizing Flows** Making the normalizing flows longer (Config F, two times the affine coupling blocks, with less feature maps to keep the parameter count constant) has a small negative impact on the FID. The model hence does not seem to be limited by the length of the normalizing flows (width of the model), but rather by the inductive bias of how the image is encoded into the hierarchical $\boldsymbol{y}$.

## 3.3 Other datasets

We train our model also using the LSUN churches and bedrooms datasets at $128 \times 128$ resolution. For comparison we also train a Glow-like model from scratch using similar capacity and the same computational resources as was used to train our own model. From Figure 7b we see that our model consistently yields much better FID than Glow. We did not experiment tuning the parameters of our model with the lower-resolution data. Because of the limitation in the parameter budget, the Glow-like model is not as deep as in Kingma & Dhariwal (2018). Uncurated samples from the models described above are found in Appendix C.

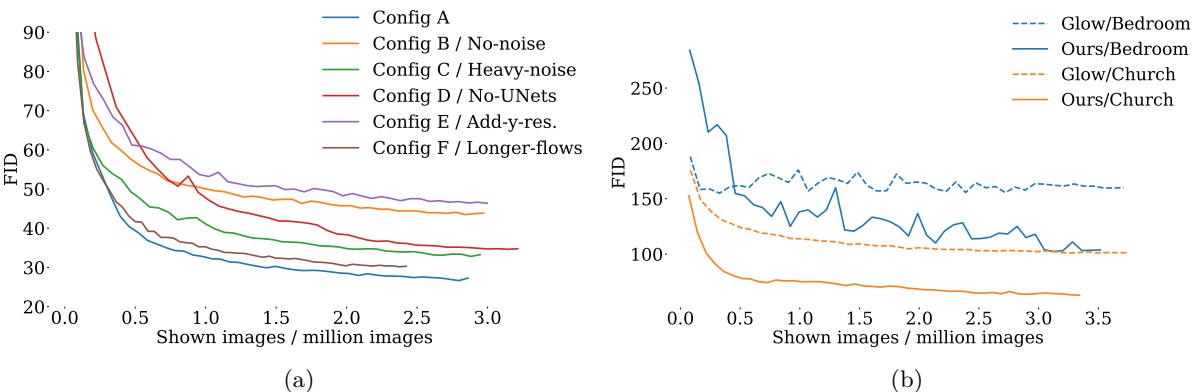

(a)                      (b)

Figure 7: (a) Ablations with CelebA-HQ $256 \times 256$. Effect of model strucure and parameters on the achieved FID measured during training. The FID is computed using 25k samples with all the training data-augmentations enabled. Each model has the same learning rate and approximately the same number of parameters. A proper level of added noise to $y$-variables improves the FID drastically. Addition of U-Nets into the flows also has a large effect. Interestingly, the FID is also very sensitive to the number of resolutions modeled with flows in the hierarchy. (b) FIDs during the training of our model on LSUN church and LSUN bedroom at $128 \times 128$-resolution, compared against a Glow-like model with similar capacity. The FID is computed using 25000 random samples from the entire trainset (which is $> 3$ million images for LSUN bedrooms), causing the noise in the measurements.

## 4 Discussion

Flow-based models have many useful properties, but have suffered from poor sample quality relative to many other families of generative models. While not on par with the current best GAN and diffusion models, the exactly invertible generative model presented in this work yields much higher-quality samples than the previous state-of-the-art invertible models. Moreover, we have shown that by constructing a hierarchical stack of conditional normalizing flows we can separate high and low-level features and model them conditionally on each other to resolve some of the issues concerning standard normalizing flow models. We find that data augmentation or regularization with noise is essential for our model to perform well. Studying different regularization methods and their effect on the latent decomposition as well as the applicability of the latent space of our model to downstream tasks would be interesting avenues for future research.

**Limitations.** Our model has a relatively large number of hyperparameters, such as the noise levels $\alpha$, that drastically affect its performance. While we have presented empirical evidence why certain choices might be better that others, we have no principled method of optimizing those values. Adding them as optimization parameters and employing the VAE loss is hardly an option, since the VAE loss also requires complex parameter-tuning for good sample quality. While our model yields better samples than other flow models, they still lag behind GANs and DDPMs. Like other flow models, ours also has difficulties modeling high-resolution, highly variable data, as seen in the LSUN results. Finally, our model does not directly extend to conditional tasks like inpainting or denoising in a principled Bayesian way due to the model not yielding an exact likelihood but only a bound.

### Acknowledgments

We thank Pauli Kemppinen and Erik Härkönen for help with the code release. This work was partially supported by the European Research Council (ERC Consolidator Grant 866435), and made use of computational resources provided by the Aalto Science-IT project.

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

## A Cumulative zeroing of the latent code for ablations

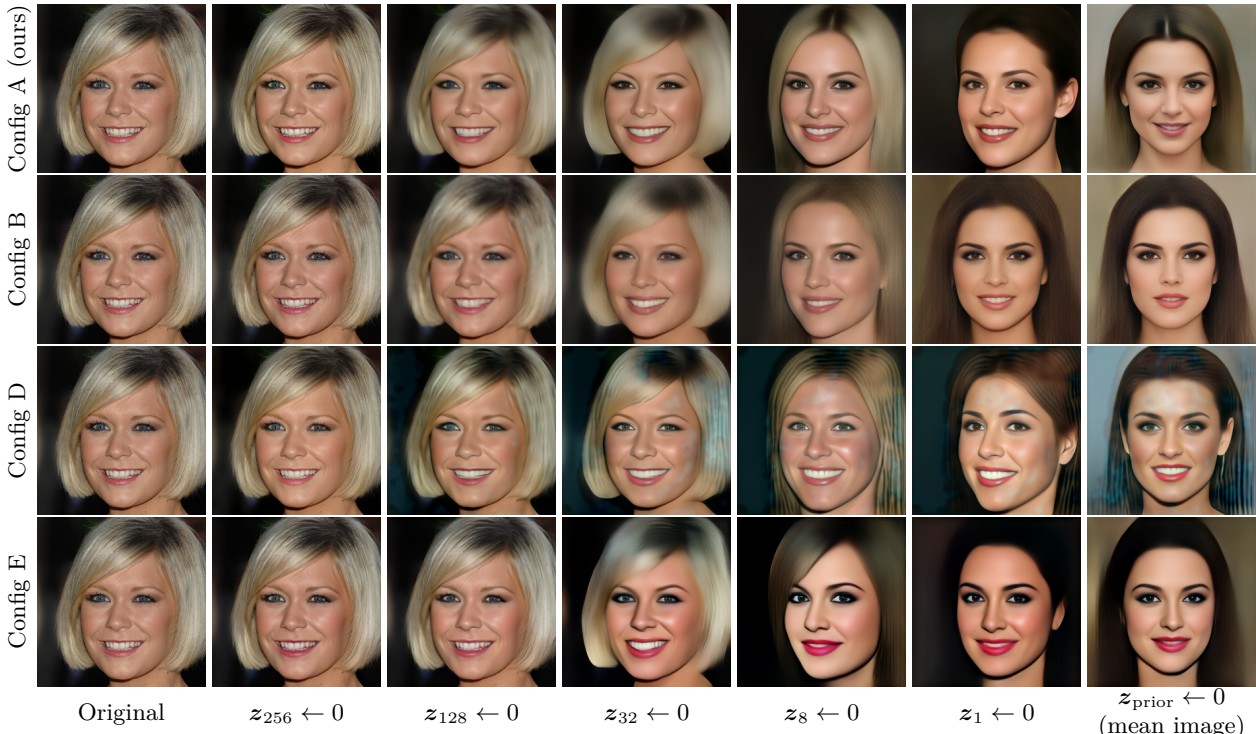

Figure 8: Cumulatively setting the latent-code of a real image to zero starting from high-resolution latents. Highlighting the failure cases of the ablations. **Config B / no noise**: Part of the low-resolution latent code is used for attempting to model hair texture while the color of the hair is lost after setting $z_8$ to zero. **Config D / No U-Nets**: Same problem as with Config B, but with a coarser hair-texture encoded to the $1 \times 1$-resolution latent. In fact, the hair texture might be from a learned constant since it is visible in the mean image as well. The $1 \times 1$ latent merely modulates this texture. Low-frequency noise-artifacts can also be clearly seen after $z_{32}$ is set to zero. **Config E / different $y$-repr.**: Note that since Config E also has $z_{64}$ and $z_{16}$, those are also cumulative set zero, between the $z_{128}$ / $z_{32}$ and $z_{32}$ / $z_8$ columns, respectively. The intermediate results are not viualized here. Very fine hair-texture is carried down to $32 \times 32$-resolution. Some high-level features such as the coloring is completely lost by the $1 \times 1$ latent.

## B    Does the model memorize the dataset?

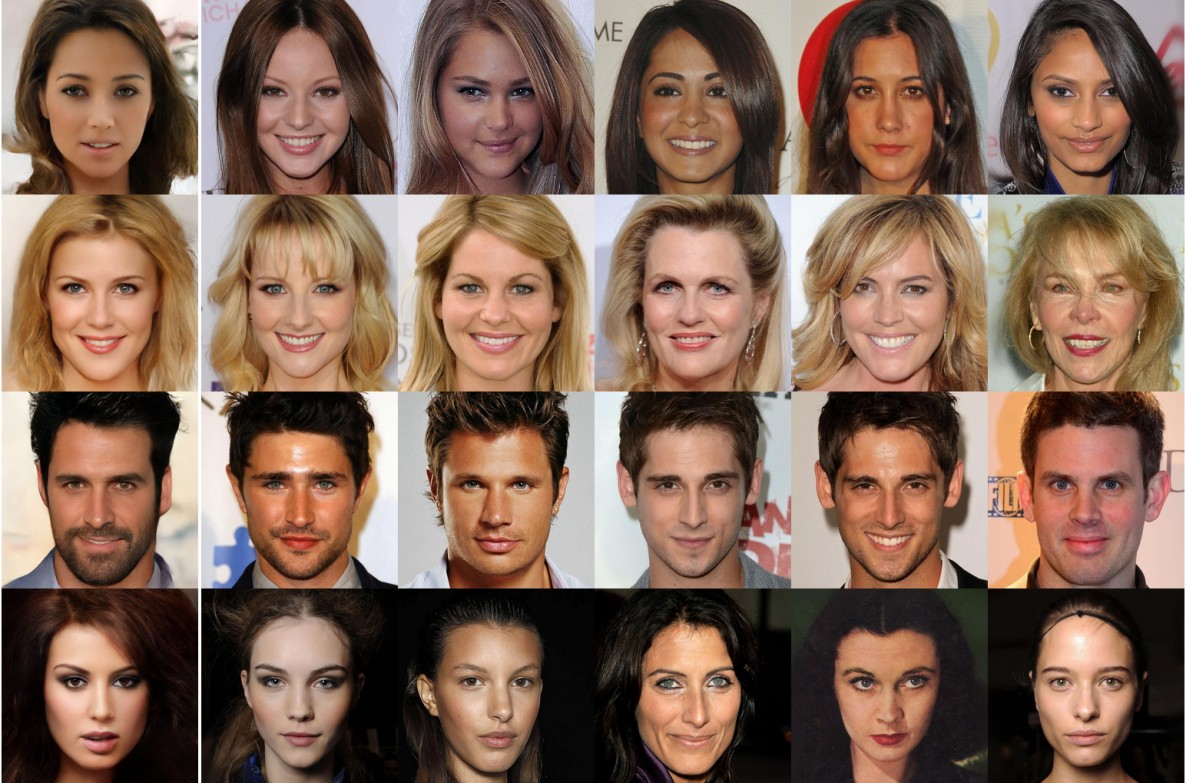

Figure 9: 5 $L_2$-closest training images (columns on the right) for generated images with Config A (rows) with the CelebA-HQ dataset. We see that the sampled images do not appear in the dataset.

## C   Comparison with a Glow with similar capacity

Ours                                                                    Glow

Figure 10: Uncurated samples from our model and from a Glow-like model with similar capacity and the same training time. All models use truncated sampling with $\sigma = 0.875$.

# D   Uncurated FFHQ $256 \times 256$ samples with model Config A

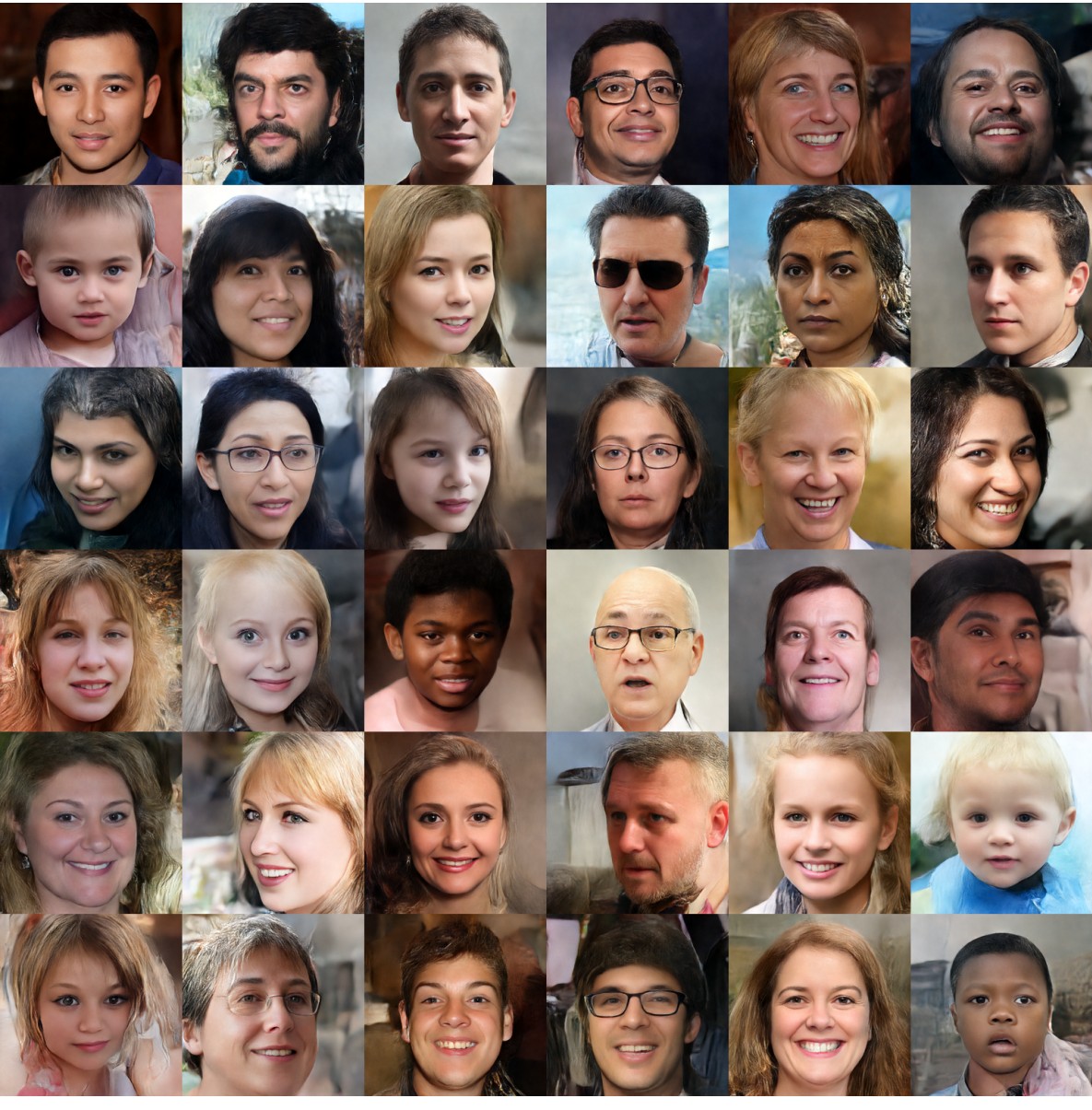

Figure 11: Uncurated samples from model with Config A trained with FFHQ $256 \times 256$. Sampled with truncation $\sigma = 0.7$ for latent resolutions larger than 1.

## E   2D Toymodel

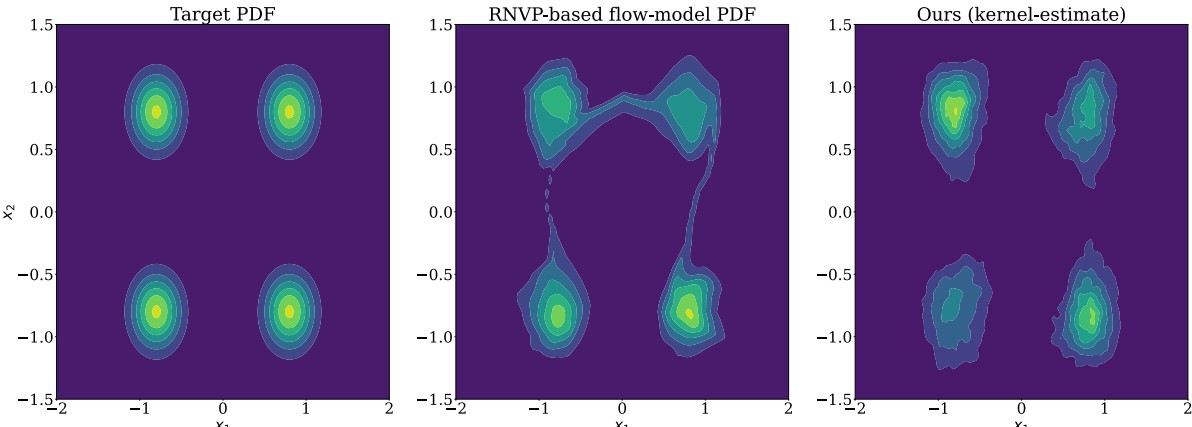

Figure 12: Real NVP-based normalizing flow (Dinh et al., 2016) and our model trained on a 2-dimensional mixture of Gaussians target distribution (left). The models have similar capacity. The standard normalizing flow (middle panel) suffers from the well-documented problem of failing to separate the modes due to the invertibility constraint of the architecture. While our model (right panel) does not capture the relative weights of the modes correctly, it captures the multi-modality better than the reference. The dimensionality of the latent $\boldsymbol{y}$ in our model is $\dim(\boldsymbol{y}) = 1$, modeled by a neural spline flow (Durkan et al., 2019) as the prior $f_{\mathrm{prior}}$. In this experiment $f_{\mathrm{cond}}$ is a conditional real NVP and the encoder and decoder are small fully-connected neural networks.

## F    Detailed network architecture

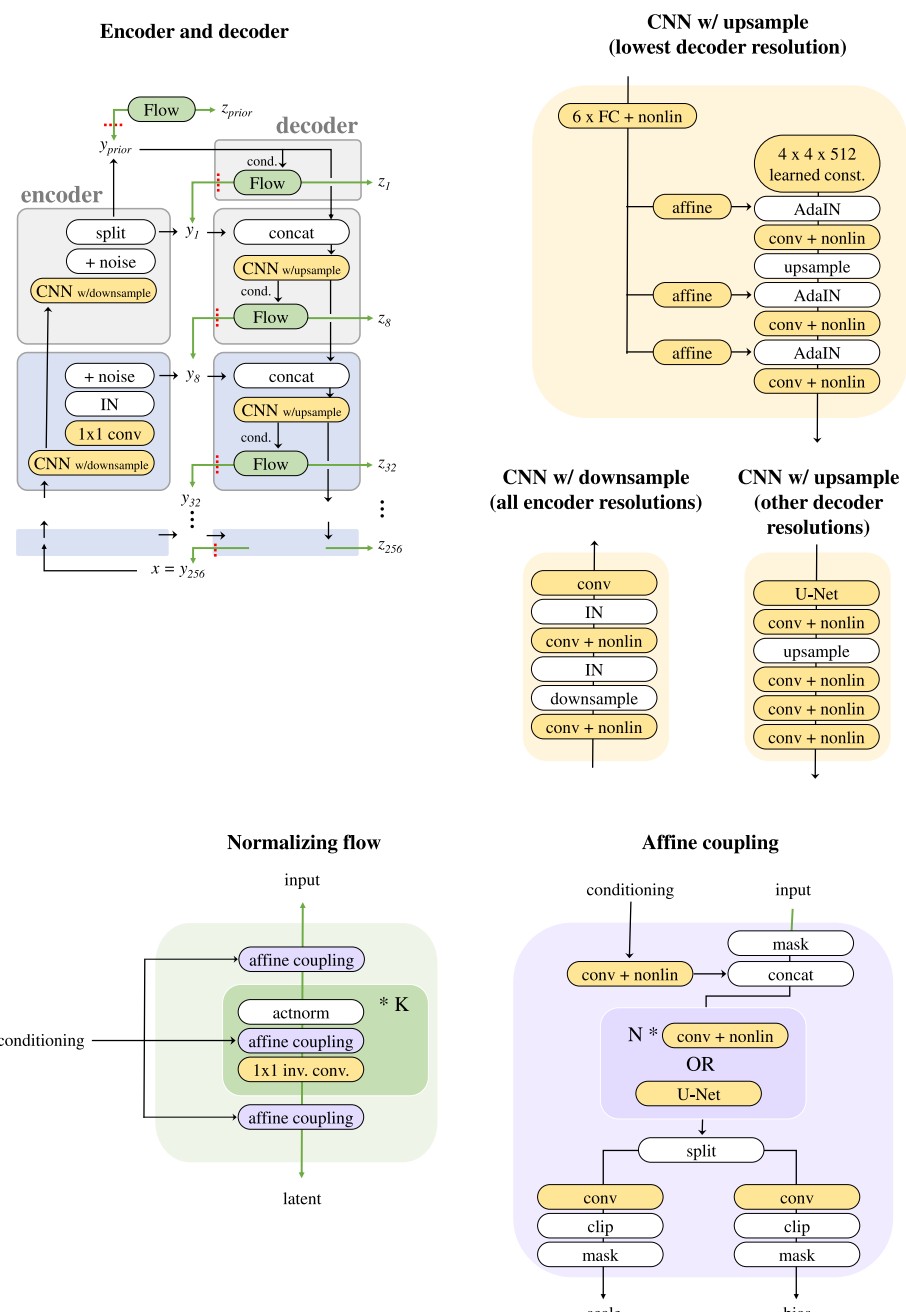

Figure 13: Detailed architecture. IN denotes instance normalization, and also appears in the U-Nets and among the affine coupling convolutions. The non-linearities are leakyReLUs apart from the conditioning of checkerboard-masked U-Net -type affine coupling blocks and the first of the $N$ convolutions of convolutional-type affine coupling blocks. Multiple units of CNN up/downsample -blocks are concatenated if there is a change of resolution differing from 2 (e.g. from $\boldsymbol{y}_{32}$ to $\boldsymbol{y}_8$). In case of this stacking, the U-Net at the beginning of the CNN-upscaler is omitted from blocks other than the first. The last and first affine coupling-layers of a normalizing flow-block use U-Nets (apart from $\boldsymbol{y}_8$ and $\boldsymbol{y}_{256}$) and completely mask out the flow-input. That is, the scales and biases are computed only using the conditioning signal.

## G    Additional metrics

Table 1: Negative log-likelihood (NLL, lower values are better) from our model and Glow. The likelihood-bound for our model is computed as the VAE ELBO as discussed in Section 2.2, which is not directly the optimization target of our model, partially explaining the performance difference with Glow, which directly optimizes for likelihood. All values are measured by us using our own implementations apart from Glow / CelebaHQ-256, which is taken from Kingma & Dhariwal (2018). As our focus is on improving FID, which is not necessarily computed against a specific test-set, we do not have a separate test-set and all our values are computed against the train-set.

| Model / dataset | NLL / bits-per-dimension |
|---|---|
| Ours / CelabaHQ 256 (5bit) | $\leq 1.3$ |
| Glow / CelabaHQ 256 (5bit) | $= 1.03$ |
| Ours / LSUN-Church 128 (8bit) | $\leq 4.0$ |
| Glow / LSUN-Church 128 (8bit) | $= 3.6$ |
| Ours / LSUN-Bedroom 128 (8bit) | $\leq 3.8$ |
| Glow / LSUN-Bedroom 128 (8bit) | $= 3.4$ |

## H    Hyperparameters and training details

Table 2: Training details for Config A with CelebA-HQ/FFHQ $256 \times 256$

| Name | Values |
|---|---|
| Batch size | 16 |
| Batch size Var($\boldsymbol{X}$) | 4 |
| Optimizer | Adam (Kingma & Ba, 2014) with $\beta_1 = 0.9$, $\beta_2 = 0.999$ |
| LR (encoders/decoders/flows) | $5 \times 10^{-4}$, $2 \times 10^{-3}$, $5 \times 10^{-3}$ |
| LR decay | Multiplicative (encoders / decoders+flows) 0.92/0.95 |
| Encoder parameter freeze at | 60 epochs |
| Gradient $L_2$ clipping | 50.0 |
| # GPUs | $2\times$ V100 16 GB |
| Train time | 96 h |
| Total parameter count | 80.15 M |

Table 3: Training details for Config A (LSUN) with LSUN church / bedroom $128 \times 128$

| Name | Values |
|---|---|
| Batch size | 16 |
| Batch size Var($\boldsymbol{X}$) | 8 |
| Optimizer | Adam with $\beta_1 = 0.9$, $\beta_2 = 0.999$ |
| LR (encoders/decoders/flows) | $3 \times 10^{-4}$, $1 \times 10^{-3}$, $3 \times 10^{-3}$ |
| LR decay | Multiplicative (encoders / decoders+flows) 0.92/0.95 |
| Encoder parameter freeze at | 60 epochs |
| Gradient $L_2$ clipping | 50.0 |
| # GPUs | $1\times$ V100 16 GB |
| Train time | 96 h |
| Total parameter count | 74.58 M |

**Data Preprocessing**    We augment each dataset by adding uniform 1/255 noise to 8-bit images normalized to $[0, 1]$ on top of which we also add slight zero-mean Gaussian noise with standard deviation $5\times10^{-3}$. During training, we apply random horizontal flips with probability $p = 0.5$.

**FID Measurement**    When comparing to Glow, we compute the FID using 30000 samples (the full CelebA-HQ -dataset). We use 5-bit dequantization (and the tiny Gaussian noise-augmentation mentioned in the previous paragraph), when computing the value for Glow, with the result being a few points weaker (56.8) for 8-bit data. We use truncated sampling with $\sigma = 0.8$ when generating images with Glow. Our model uses 8-bit images.

Table 4: Model hyperparameters for Config A. The LSUN-128 models are trained with the same configuration, but the Flows at resolutions -parameter ($R_i$) set to [128,64,32,8,1, prior] instead. The affine-coupling split type uses format $M \times$ split type, where $C$ denotes splits along the channel dimension and $S_K$ a spatial checkerboard split with $K$-pixel alternation. Coupling types are listed starting from the side of the input (e.g. the channel-splits are in general closer to the input than the latent). Affine coupling blocks with spatial splits (denoted with $S_K$) use U-Nets as their forward neural networks. There is no additional source of noise for the flow at the highest resolution $\boldsymbol{x} = \boldsymbol{y}_{256}$ and hence $\alpha$ is not defined there. LeakyReLU-nonlinearities use slope 0.1.

| Name | Values |
|---|---|
| Flows at resolutions ($R_i$) | [256, 128, 32, 8, 1, prior] |
| Number of channels at flows | [3, 4, 8, 8, 408, 4] |
| Noise scale to flow ($\alpha$) | [n/a, 0.4, 0.05, 0.05, 0.05, 0.05] |
| Noise scale to decoder ($\alpha$) | [n/a, 0.4, 0.05, 0.05, 0.075, 0.075] |
| Flow-loss-weight ($w_i$) | [1/(256x256x3), 10/(128x128x4), 10/(32x32x8), $1/2^9$, $1/2^9$, $1/2^9$] |
| Flow-lengths ($K$) | [4, 8, 8, 8, 8, 8] |
| Affine-coupling-split-type | $[2C, 2S_2], [2C, 2S_2, 2S_4, 2S_8], [2C, 2S_2, 2S_4, 2S_8]$ , $[8C], [8C], [8C]$ |
| Affine-coupling-length ($N$) | [4, 4, 4, 4, 4, 4] |
| $1 \times 1$ invertible convolution kernel | [free-form, free-form, free-form, free-form, unitary, unitary] |
| Total latent space dimensionality $|\boldsymbol{y}|$ | 271360 |
| Affine-coupling conditioning channels | [16, 32, 64, 128] |
| Affine-coupling hidden layer channels | [32, 64, 128, 128] |
| Encoder hidden layer channels | [64, 256, 256, 512] |
| Decoder hidden layer channels | [64, 256, 512, 512] |

Table 5: Model hyperparameters for our reference Glow implementation using notation of Kingma & Dhariwal (2018). We use the Adamax variant of Adam for optimization, with learning rate $5 \times 10^{-3}$ and batch size 16. The data is dequantized to 8 bits. Gradient magnitude is clipped at 50.0.

| Name | Values |
|---|---|
| Levels ($L$) | 5 |
| Depth per level ($K$) | 24 |
| Coupling type | Additive |
| Hidden channels coupling layers | 256 |

