# OpenReview forum: "Invertible Hierarchical Generative Model for Images"
_TMLR — Accepted by TMLR_

### Review · Reviewer_WMXD · 2023-07-19

**Summary Of Contributions:**

This paper proposes a novel flow-based hierarchical generative model. It works by organizing multiple shallow conditional normalizing flows in a hierarchy. A deterministic encoder (with added noise) computes the conditioning information for each level of the hierarchy. The decoder leverages conditioning normalizing flows at each “layer” to generate the image. A learned prior (also a flow-based model) models the aggregate posterior for sampling the initial latent.

The proposed model mainly presents improvements compared to GLOW (Kingma & Dhariwal, 2018) in terms of FID. It is also shown how the hierarchical structure of the model is better able to capture variations to the generated images at multiple levels of abstraction. The proposed model is quite complex, and a hyper-parameter analysis reveals that it is also quite sensitive to some of the hyperparameter choices, especially the amount of encoder noise added. Though once properly tuned the model performs is able to improve substantially compared to GLOW.


**Audience:**

Yes

**Broader Impact Concerns:**

N/A.

**Claims And Evidence:**

Yes

**Requested Changes:**

I think the paper is sufficiently polished and ready for publication, though I have listed some suggestions for improving this work in the comments above.

**Strengths And Weaknesses:**

+ This is a well written paper that is relatively easy to follow. The clarity can be improved further by adding an algorithm box for sampling and inference in the model, though this is minor.

+ The experimental evaluation and hyper-parameter analysis is sufficiently thorough, and it is clear that this approach works and can improve over GLOW.

+ I appreciate that the authors are clear about the current limitations of this work in Section 4, which is important.

- The main improvement of the proposed is to GLOW, which is 5 years old and hasn’t been SOTA for image generation purposes in a long time. However, for the purposes of closing the gap of flow-based to diffusion/gan-based models, this work is certainly valuable.

- The model is quite complex and sensitive to hyper-parameter values (for which no good heuristics are available), though the authors clearly acknowledge this.

- No literature from 2023 is covered and only a single paper from 2022. While I am not very familiar with the latest state of flow-based generative models, it would make sense to me to try include a discussion of more recent advances in generative (flow-based) models more generally.

 - The main selling point of using a flow-based model seems to be the ability to perform exact inference and compute exact likelihoods. However, the paper is mainly concerned with improvements in terms of FID. While this is understandable, as the comparison is to another flow-based model, it would have also made sense to include some results that better demonstrate the benefit of flow-based models compared to diffusion/gan-based models even though those are not directly compared to.

---

> ### Author Response · Authors · 2023-09-06
> **Response to review**
>
> We are very thankful to the reviewer for their time and for providing useful input in improving the manuscript. We have updated the paper accordingly.
>
> > The clarity can be improved further by adding an algorithm box for sampling and inference in the model.
>
> We thank the reviewer for this remark. We have added an algorithm-box to clarify the inference and sampling procedures.
>
> > Literature coverage and recent advantages flow-based generative models
>
> To the best of our knowledge there is not much literature from 2022 or 2023 available regarding flow-based models in a general setting. Most work tends to focus on specific applications of normalizing flows on domains like physics, computer vision or general inverse problems.
> Regarding flow-based models in general, there has been some work on building more efficient invertible layers for normalizing flows [1,2]. Furthermore, continuous-time normalizing flows (neural ODEs) have had some interesting work recently [3,4,5], building a link between flow-models and diffusion models with elements of optimal transport. We would be happy to cite these works and applications.
>
> [1] https://proceedings.mlr.press/v162/meng22a.html (ButterflyFlow: Building Invertible Layers with Butterfly Matrices)
>
> [2] https://arxiv.org/abs/2301.09266 (FInC Flow: Fast and Invertible k×k Convolutions for Normalizing Flows)
>
> [3] https://arxiv.org/abs/2210.02747 (Flow Matching for Generative Modeling)
>
> [4] https://arxiv.org/abs/2209.03003 (Flow Straight and Fast: Learning to Generate and Transfer Data with Rectified Flow)
>
> [5] https://arxiv.org/abs/2302.00482 (Improving and Generalizing Flow-based Generative Models with Minibatch Optimal Transport)
>
> > ...it would have also made sense to include some results that better demonstrate the benefit of flow-based models compared to diffusion/gan-based models even though those are not directly compared to.
>
> Some benefits of normalizing flows we discuss in the paper include fast and efficient sampling (when compared with diffusion models), exact inference (GANs need a separate inference network) and perfect invertibility (GANs, diffusion models with stochastic encoders fail to reconstruct encoded images). Sampling speed and reconstruction error can be condensed into a few numbers out of which reconstruction error is by construction 0. For completeness, we can add a measurement of sampling speed.

---

### Review · Reviewer_ugi5 · 2023-07-25

**Summary Of Contributions:**

This paper proposes a new invertible generative model using hierarchical normalizing flows. The U-Net-based encoder-decoder model creates an intermediate representation $\boldsymbol{y}$, which has fine-to-coarse spatial hierarchical structures. Then, the model transforms an image and intermediate representations into latent representations $\boldsymbol{z}$ using invertible flow models at each level. This paper applies the proposed method to the CelebA-HQ 256x256 and LSUN datasets and quantitatively evaluates its image generation performance using FID. For CelebA-HQ 256x256, the proposed method is quantitatively evaluated by visualizing generated images and latent representations.

**Audience:**

Yes

**Broader Impact Concerns:**

This paper does not explicitly mention Broader Impact. However, similarly to other image generation technology,  we may carefully consider the social negative impacts of its misuse, such as fake images.

**Claims And Evidence:**

Yes

**Requested Changes:**

- P.3, Eq. (3): *... where $p\_\theta$ is the distribution induced by a conditional normalizing flow $\boldsymbol{z}\_{\text{cond}} = f\_{\text{cond}}(\boldsymbol{x}; \boldsymbol{y}, \theta)$ with a unit Gaussian prior.*: It may not be easy to understand this part mathematically. Is $p\_\theta$ a probability distribution constructed by push-forwarding $\boldsymbol{z}\_{\text{cond}}\sim \mathcal{N}(0, I)$ using $f^{-1}\_{\text{cond}}(\cdot; \boldsymbol{y}, \theta)$
- P.3, Eq. (4): Similarly, is $p_\varphi$ a probability distribution constructed by push-forwarding $\boldsymbol{z}\_{\text{prior}}\sim \mathcal{N}(0, I)$ using $f^{-1}\_{\text{prior}}$?
- P.4: [...] we optimize minimize [...] -> [...] we optimize to minimize [...]
- P.5: The noise can be though as [...] -> The noise can be thought of as [...]
- P.8, Figure 4: What exactly does the term "truncated sampling" used in image generation (e.g., Figure 4) refer to? Also, how does the parameter $\sigma$ work there?

**Strengths And Weaknesses:**

**Claim and Evidence**

If I understand correctly, this paper points out four problems of Glow and proposes a method to remedy them:
1. difficult to model multi-modal distribution
2. aliasing artifacts in generated images
3. limited variations of generated images
4. uneven contribution of decomposed latent representations

- For the first problem, if I get all the information, this paper does not verify it, at least directly. However, I think the fact that FID scores are improved in image generation tasks on two image datasets (CelebA-HQ 256x256 and LSUN) can be considered as evidence to support this claim.
- For the second problem, Figure 1 qualitatively evaluates the improvement.
- For the third and fourth problems, Figures 5 and 6 evaluate the effect of $\boldsymbol{z}$'s changes on generated images. Although both are qualitative evaluations of a single image, they are consistent with their claims, at least for the example presented.


**Audience**

The Flow-based model is one of the most promising image generation models, along with the score-based, GAN-based, and Transformer-based models. Its improvement is an important issue since the aliasing problem of Glow is directly related to the impression of the generated images. Therefore, this paper should be of interest to TMLR readers.


**Weaknesses**

The proposed method certainly improves over Glow by the comparisons from various aspects. However, as pointed out in this paper, this paper does not compare the proposed models with other generative models, such as GAN-based and score-based models. In fact, according to [1], the SOTA of CelebA-HQ 256x256 FID is 3.25 of the GAN-based model, and the FID of SOTA of the score-based model is also significantly different from the proposed method.

[1] https://paperswithcode.com/sota/image-generation-on-celeba-hq-256x256

---

> ### Author Response · Authors · 2023-09-06
> **Response to review**
>
> First of all, we would like to thank the reviewer for the valuable comments!
>
> > Claims and evidence on the multimodality problem
>
> The reviewer is correct in pointing this out; we perhaps use the term “multimodal” in a slightly imprecise manner, rather meaning the general difficulties in modeling complex image distributions. We are happy to clear the language up. In the current form of the paper, there is no direct evidence in support of our model handling multimodal distributions better than the baseline Glow does.
> We have added an experiment to the revised version where we show that for a 2-dimensional toy-data testcase, our model does indeed perform better than a classical flow-model of similar capacity. We see, however, that this experiment is better relegated to the appendix as it is more supporting evidence rather than a main finding.
>
> > Comparison to GANs and score-based methods
>
> We would be happy to cite these numbers and their sources in a revision.
>
> > Requested changes
>
> We have added clarifying remarks about push-forwards in context of the invertible functions $f_{cond}$ and $f_{prior}$.
> Additionally, we have fixed the typos and related language errors.
> We also have clarified that truncation means reduced-temperature sampling, which is the term used in the Glow paper for using a variance < 1 when sampling, even though the model has been trained assuming that the flow base-distributions have unit variance. Using progressively sampling smaller variance leads into smaller variance in the samples but also usually higher quality samples.

---

> > ### Comment · Reviewer_ugi5 · 2023-09-11
> >
> > I thank the authors for answering my questions. I am mostly satisfied with the authors' responses. Here are my comments on the responses:
> >
> > > Claims and evidence on the multimodality problem
> >
> > I thank the authors for providing additional experiment results using synthesis data in Appendix E. I agree that this supports the claim that the proposed method can model the multi-modal distributions.
> >
> > > Comparison to GANs and score-based methods
> >
> > *We would be happy to cite these numbers and their sources in a revision.*: I appreciate that the authors considered my comments and were willing to update the draft. However, I could not find a place that reflected this change. Let me know if I miss some information.
> >
> > > Requested changes
> >
> > OK

---

> > > ### Author Response · Authors · 2023-09-12
> > >
> > > Thank you for the response and apologies for the confusion. We will have the suggested GAN and score-based model metrics and their sources included in a future revision.

---

### Review · Reviewer_xzyC · 2023-08-31

**Summary Of Contributions:**

The authors propose a generative model that is based on conditional normalizing flows. The conditioning factor enables a multi-scale architecture where each layer in the hierarchy generates the conditioning factor of the next layer with latent space at each layer having a different spatial resolution. The model resembles hierarchical VAEs with normalizing flows for the encoder and prior distributions of the latent space and a normalizing flow in the decoder but with an alternative regularization scheme.

The authors test the suggested architecture on CelebA-HQ and LSUN churches and bedrooms in terms of FID compared to glow. More importantly, they qualitatively assess the behavior of the latent space at different resolutions showing its capability to capture finer detail at higher resolutions (a  weakness of existing flow-based models).

**Audience:**

Yes

**Broader Impact Concerns:**

no concerns

**Claims And Evidence:**

Yes

**Requested Changes:**

****** regarding the prior regularization scheme in equation 4. *****

1) Do the authors think that such a loss can have a probabilistic/ information theoretic interpretation?

2) Could a regularization coefficient achieve a better trade-off between reconstruction loss and regularization?

3) The prior portion of the model resembles prior networks [1] albeit in [1] the prior network is not trainable (hence implicitly avoiding degenerate solutions). I think the authors should connect these two concepts

4) Slightly improve writing for better comprehensibility: lacking the entropy term originating from the KL--> lacking the negative entropy term of the encoder.

5) The risk of degenerate encoders is also discussed in [2] (prior to  cited Xiao's work) and solved by beta-VAEs (beta<1). For completeness, I also think the authors should cite and comment on [2]


****** regarding the architectural depth  *****

1) I think the authors at this point should also report number of layers (flows) in the architecture and number of transformations per flow for a fair comparison with layers in the VAEs. report of trainable architectures for both model categories might their argument stronger.
2) there are actually more recent deep vaes with shallower architectures [3] which are also not commented [3].


***** notational issues *****

1) there seems to be a contradiction in the notation used:

right above equation 4:

$z_{prior}=f_{prior}(y;\phi)$

in equation 7, they define $z_{prior}$ as the inverse of f_{prior}

2) minor: in section 2.2. The noise can be though**t**

References

[1] Osband, I., Aslanides, J., and Cassirer, A. (2018). Randomized prior functions for deep reinforcement learning. In Bengio, S., Wallach, H., Larochelle, H., Grauman, K., Cesa-Bianchi, N., and Garnett, R., editors, Advances in Neural Information Processing Systems 31, pages 8617–8629. CurranAssociates, Inc.]

[2]Hoffman MD, Riquelme C, Johnson MJ. The β-vae’s implicit prior. InWorkshop on Bayesian Deep Learning, NIPS 2017 (pp. 1-5).

[3] Apostolopoulou I, Char I, Rosenfeld E, Dubrawski A. Deep Attentive Variational Inference. InInternational Conference on Learning Representations 2021 Oct 6.

**Strengths And Weaknesses:**

strengths:

1) The authors improve the sampling quality of flow-based generative models.

2) They demonstrate qualitatively that i) lower resolution layers are capable of capturing global features of the image. ii) higher resolution stochastic layers do still yield variations in the image at finer details. this is in contrast to glow models where these layers do not affect the sampled images.

weaknesses:

1) albeit the proposed model outperforms the glow baseline, it might be good if the authors give a picture of where their model stands in terms of other generative models like vaes.

2) they should also compare with glow in terms of sampling time and model size (trainable parameters)

---

> ### Author Response · Authors · 2023-09-06
> **Response to review (1/2)**
>
> We thank the reviewer for the detailed review and providing important references to relevant previous work.
>
>
> > Model size and sampling time
>
> Although not directly reported in the original Glow paper, inspecting the official pre-trained model yields around 220 M parameters for Glow. Our model has around 80 M parameters in total, which is reported in Appendix G of the initial submission. We have also added a mention about this in the main text in the revision. For completeness, we can add a measurement of sampling speed.
>
>
> > The risk of degenerate decoders and the work of Hoffman et al.
>
> We are thankful for the reviewer having brought this to our attention. We have added a note about this work to the revision.
> The degeneracy discussed in Hoffman et al. is very much related but slightly weaker from what Xiao et al. and we have observed. In Hoffman et al. the degeneracy is only with respect to the variance of each $q(z|x)$ (which can then be fixed with the regularization-term as the reference suggests). In our case, since the prior is modeled by a normalizing flow (i.e. it is not just a static Gaussian prior) that can have non-zero log Jacobian determinant there is also pressure for $r(z)$ (as defined in Hoffman et al.) to have zero mean as it is in the interest of the prior flow to try to expand the probability mass of $r(z)$ (= positive log Jacobian determinant) to match the Gaussian base distribution of the prior flow.
>
> > Number of layers and transformations
>
> The number of layers of the various normalizing flows, including the number of parameters, is reported in Appendix G of the initial submission. There is also a brief, although not very precise, mention about deep-VAE depth and the depth of our model in the paragraph “Trading depth for width”. We are happy to provide more detail if requested.
>
> > Deep VAEs with shallower architectures (Apostolopoulou et al.)
>
> We thank the reviewer for bringing this work to our attention, we have updated the paper with the reference. The authors of Apostolopoulou et al. have made the same conclusion that very large depth in likelihood-based generative models is not necessary if long-distance inter-dependencies within the latent are considered with more care. We have updated our paper with the proper reference. Our work however, has a slightly different goal since Apostolopoulou et al. seem to be more interested in improvements in the ELBO and not image quality and FID. They also work only with relatively small 32x32 images.
>
> > Notational issues and writing comprehensibility
>
> We have corrected these in the revision.
>
>
> > The prior portion of the model resembles prior networks of Osband et al.
>
> As far as we understand, Osband et al. suggest to
>
> 1. Instead of directly optimizing a deep Q network from episodical data using a TD-loss, instead optimize a deep Q network that has been perturbed by a random (but fixed) prior function p.
> 2. Train and return an ensemble of these perturbed deep Q networks.
> The justification of the prior is to provide "motivation" or "curiosity", that is, to bias the agent to take certain actions, in settings with sparse rewards to avoid completely random searches that might never lead to any reward.
>
> We see our model more from the point of view of [1] and [2], the references 35 and 65 in the work of Osband et al., where the prior of theta (which are parameters for a deep Q network) is fitted with the tools of Variational Inference. The underlying idea seems nevertheless the same as in the given reference: bias the agent’s exploration so that it is more efficient than more classical methods for controlling the exploration-exploitation tradeoff.
> The difference is that instead of fitting the posterior variational approximation $q(\theta|data)$ with a parametric $p(\theta)$, we would fit the aggregate $q(\theta)$ with $p(\theta)$ in a separate step.
> Due to these small differences we believe additional experimentation is required on the application of our model to a reinforcement learning setting before any conclusions can be made. We think it is an interesting avenue for future research.
> We would be, regardless, happy to point out the similarity.
>
>
> [1] https://ojs.aaai.org/index.php/AAAI/article/download/11946/11805 (BBQ-Networks: Efficient Exploration in Deep Reinforcement Learning for Task-Oriented Dialogue Systems)
>
> [2] https://arxiv.org/pdf/1711.11225.pdf (Variational Deep Q Network)

---

> ### Author Response · Authors · 2023-09-06
> **Response to review (2/2)**
>
> > How does the model stand in terms of other generative models like VAEs?
>
> We have discussion and comparisons to other types generatives models in the introduction and related work sections of the manuscript. We also see that the relation to VAEs is strongly related to the reviewers questions about the regularization scheme and probabilistic interpretation of our model. We hope that our answers to those questions also answer to this question.
>
>
> > Do the authors think that such a loss can have a probabilistic/ information theoretic interpretation?
>
> In a sense, the loss can be seen as the beta-VAE [1] -loss but with $\beta=0$, meaning that we only care how well the original data x can be inferred with z. We only implicitly bottleneck the encoder–decoder system via choosing the dimensionality of y to be much smaller than that of x and adding noise whose magnitude is given by a hyperparameter. We then separately model the aggregate distribution of y with the “prior” normalizing flow.
>
> > Could a regularization coefficient achieve a better trade-off between reconstruction loss and regularization?
>
> As our model always performs exact reconstruction up to floating point precision, there is no explicit trade-off between reconstruction and regularization in the traditional sense of VAEs. While full information about a datapoint is not encoded into the latent y, it can nevertheless be reconstructed via the conditioning mechanism of the conditional flow $f_{cond}$ (Figure 2).
> The noise level hyperparameters $\alpha$ of our model do have a regularizing effect. With large values for $\alpha$ the latent y is almost entirely noise and carries little information about the original x. In this case the conditional flow $f_{cond}$ reduces into essentially an unconditional model, since the conditioning is almost entirely noise which is best ignored by the model. In the other extreme with small values of $\alpha$ y will be maximally informative about x, but the aggregate posterior will be a collection of zero-variance spikes ($q(z|x)$ is deterministic). This might lead to overfitting in the decoder which during sampling may result in poor samples as the prior flow is likely to yield out-of-distribution samples from the perspective of the decoder.
>
>
> [1] https://openreview.net/pdf?id=Sy2fzU9gl (beta-VAE: Learning Basic Visual Concepts with a Constrained Variational Framework)

---

### Decision · Action_Editor_MmvZ · 2023-10-17

**Recommendation:** Accept as is

**Comment:**

This paper proposes an interesting new invertible generative model using hierarchical normalizing flows and demonstrates performance improvements of the suggested approach over Glow.  All reviewers were positive about the paper and recommended acceptance.

As there were not any notable issues outstanding after the rebuttal, I am happy to recommend the acceptance of the paper "as is".  I would still encourage the authors to add the sampling times requested by Reviewer xzyC in the final version.

**Audience:**

This work is of clear relevance to the TMLR audience and I believe many in the community will be interested in the work.  The reviewers were again unanimous that the paper fills TMLR's acceptance criteria on this matter.

**Claims And Evidence:**

All reviewers agree that the paper convincingly supports its claims with clear evidence.  No major technical issues were raised and reviewers felt the paper does a good job of making the limitations of the paper clear.